# Do Patients with Complaints Attributed to Chemicals in the Environment Trust in Biomonitoring as a Valid Diagnostic Tool? A Prospective, Observational Study from a German University Outpatient Clinic

**DOI:** 10.3390/ijerph22071143

**Published:** 2025-07-18

**Authors:** Claudia Schultz, Catharina Sadaghiani, Stefan Schmidt, Roman Huber, Vanessa M. Eichel

**Affiliations:** 1Practice for General Medicine, 79194 Gundelfingen, Germany; 2Institute for Infection Prevention and Hospital Hygiene, Medical Center—Medical Faculty, University of Freiburg, 79106 Freiburg, Germany; 3Department of Psychosomatic Medicine and Psychotherapy, Medical Center—Medical Faculty, University of Freiburg, 79106 Freiburg, Germany; 4Center for Complementary Medicine, Medical Center—Medical Faculty, University of Freiburg, 79106 Freiburg, Germany; roman.huber@uniklinik-freiburg.de

**Keywords:** multiple chemical sensitivity (MCS), environmental medicine, biomonitoring, chemicals, environment

## Abstract

Biomonitoring often yields normal results in patients who report environmental sensitivities, such as in multiple chemical sensitivity. This study examined whether biomonitoring results influence disease attribution and perception. Patients over 18 presenting for the first time to the University Environmental Medicine Outpatient Clinic in Freiburg with suspected complaints linked to heavy metals, wood preservatives, pesticides, solvents, or mold spores were included. Illness perceptions were assessed before and after biomonitoring using the Illness Perception Questionnaire (IPQ-R). Of 358 patients, 51 met inclusion criteria; 3 showed relevant findings, and 15 did not attribute their symptoms to environmental causes at baseline. The remaining 33 patients were analyzed. After receiving a normal biomonitoring result, only seven patients (21%) altered their illness attribution. These individuals also reported milder perceived consequences, less personal control over the illness, and showed lower levels of somatization and compulsiveness than those who maintained their original attribution. Most patients remained convinced of an environmental cause despite unremarkable findings. This suggests that a substantial subset of patients is strongly attached to an environmental explanation for their symptoms, with stable attribution linked to higher psychological symptom burden and a belief in personal control over the illness.

## 1. Introduction

Rapid industrialization and the proliferation of synthetic chemicals over the past century have led to widespread low-level environmental exposures, raising concerns about their potential effects on human health. Biomonitoring—the direct measurement of chemicals or their metabolites in biological samples—has emerged as a key tool for assessing internal exposure and guiding both individual risk communication and public-health interventions. Individuals seen in an Environmental Medicine Outpatient Clinic commonly report a wide range of non-specific complaints, such as fatigue, dizziness, concentration difficulties, mucosal gastrointestinal or respiratory irritations that are often attributed to heavy metals and/or multiple chemicals such as vapors from buildings, furniture or carpets, fragrances, cleaning agents, or printer emissions.

Yet, only about 15% of these cases reveal to biologically verifiable exposures; in the vast majority of patients, other diseases such as somatoform disorders (most common), skin diseases, respiratory diseases or diseases of the gastrointestinal tract are diagnosed [1]. Two syndromes are particularly relevant: multiple chemical sensitivity (MCS ICD-10 T78.4) and sick building syndrome (SBS). MCS is defined by chronic, reproducible symptoms triggered by low-dose exposure to chemically diverse agents—responses not explained by classical toxicology or allergy—and involving multiple organ systems [2]. In contrast, SBS describes symptoms (e.g., headache, mucosal irritation, fatigue) that are temporally associated with time spent in a specific building, often improving upon leaving that environment, and usually occurring in several individuals exposed to the building.

Diagnostic criteria for MCS vary internationally, according to the consensus criteria (Bartha et al., 1999), MCS is characterized by the following [2]:the symptoms are reproducible with repeated chemical exposuresthe condition is chronicsymptoms are triggered by low doses of exposure that are generally tolerated by others or were tolerated before the onset of the conditionsymptoms improve or disappear completely when triggers are avoided or removedsymptoms are triggered by different, chemically unrelated substancesseveral organs or organ systems are affected.

Causes and pathophysiology of MCS are currently discussed; the main etiological theories are post-traumatic stress disorder (PTSD), panic disorder, classical conditioning, neurogenic inflammation, limbic system dysfunction, immune system dysregulation (allergy) and oxidative stress [3,4]. A recent study on Japanese school children found the following risk factors for the development of MCS: female sex, having various allergic conditions, relocating into a new home once or more, having experienced home renovations or extensions, living near a freeway/national highway/factory/rubbish dump/source of offensive odors, not exercising, having cold hands and feet, being fatigued, having a bedtime earlier than 11 p.m., and moderate–frequent subjective stress [5].

To meet the complexity of this disease, a biopsychosocial model for a comprehensive understanding, emphasizing the interplay between psychological stressors, biological vulnerabilities, and social influences was suggested. This approach acknowledges the multifactorial nature of MCS, where psychological stressors (e.g., anxiety or traumatic experiences) may lead to and intersect with physiological sensitivities (e.g., sensitization of TRP receptor, increased oxidative stress [6]) and environmental factors (e.g., societal awareness of toxins) [7].

Many patients with suspected environmental sensitivities have a long path of suffering, a considerable reduction in quality of life and high personal costs due to alternative practitioners or individual medical health services. It is not uncommon for tests for environmental pollutants to be carried out using unsuitable methods, with the result that patients’ disease attributions are confirmed by falsely biologically verified exposures. In a follow-up examination of allegedly conspicuous exposure to pollutants (15% of a cohort of 653 consecutive patients) by an University Environmental Medicine Outpatient Clinic, the conspicuous findings could not be confirmed in any case [8].

In the present study, we investigated whether the exclusion of biologically verifiable exposures by biomonitoring has an influence on the patients’ disease model and quality of life. Drawing on our experience at the University Environmental Medicine Outpatient Clinic of Freiburg, we identified two distinct patient response profiles to biomonitoring: those whose anxiety is relieved by the targeted exclusion of biologically verifiable exposures—leading them to abandon environmental noxae as the cause of their symptoms (the “changed” group)—and those whose personal disease attibution remains unchanged despite inconspicuous biomonitoring results (the “stable” group).The patients were to be examined and compared using established questionnaires assessing their personality, quality of life, state of mind and subjective disease model before and after disclosure of biomonitoring results.

## 2. Materials and Methods

A non-interventional, monocentric study was conducted. We included patients aged 18 and older who, on their first visit to our University’s Environmental Medicine Outpatient Clinic, presented with complaints suspected to arise from quantifiable environmental exposures—specifically heavy metals, wood preservatives, pesticides, organic solvents, or mold spores. The clinic is staffed by physicians with specialized training in environmental health, clinical toxicology and exposure assessment. Patients may be referred by general practitioners or specialists—or they may self-refer—when symptoms are suspected to involve environmental factors, but the service also welcomes individuals with unexplained complaints regardless of prior attribution. This multidisciplinary setting thus serves both those who arrive expecting an environmental diagnosis and those seeking a broader differential evaluation. Active recruitment was not carried out. Primary exclusion criteria were pregnancy or breastfeeding, relevant concomitant diseases such as heart failure, liver or kidney failure, tumor disease or a known psychiatric illness, previously diagnosed MCS, sick-building syndrome or chronic fatigue syndrome as well as a history of toxicologically relevant environmental pollution. These individuals were excluded to ensure that our cohort reflected individuals at their very first presentation, without prior condition-specific interventions or counseling. This “clean” baseline reduces confounding from other diseases or treatments and prevents expectation bias arising from a pre-existing diagnostic framework. Consequently, we can more accurately assess how incorporating biomonitoring data influences initial clinical reasoning and management decisions. Secondary exclusion criteria were abnormal findings in biomonitoring that could explain the complaints and a lack of environmental attribution of the complaints.

An environmental attribution was assumed if the patients had answered item C7 (“pollution/environmental toxins” as a cause attribution) of the Illness Perception Questionnaire-Revised (IPQ-R) with “true” or “completely true” [9].

The study received a positive vote from Freiburg University Ethics Committee (EK280/13) and was conducted in accordance with the Declaration of Helsinki [10]. The registration was made in the Registry Freiburg. Before inclusion in the study, the patients were informed comprehensively in oral and written form and a written declaration of consent was obtained.

### 2.1. Target Parameter

The primary target parameter was the percentage of patients whose disease model recorded using IPQ-R changed after inconspicuous biomonitoring of the suspected environmental toxicants.

### 2.2. Secondary Target Parameters Were

-personality structure, measured with the Neo-FFI questionnaire [11,12].-disease-related quality of life measured using the Euro Quality of Life (EQ) 5D scale [13].-mental state measured with the Brief Symptom Inventory (BSI) [14].

### 2.3. IPQ-R—Illness Perception Questionnaire

The German-language version of the IPQ-R comprises 64 items that can be assigned to three question complexes [15,16]: 1. symptoms of the illness (14 items), 2. ability to influence the illness, in particular through own behavior and attitudes (32 items) and 3. suspected causes (18 items). In part three, potentially disease-causing factors such as “stress and worry”, “heredity”, “pollution/environmental toxins”, “smoking” or “accident or injury” are asked on a five-point Likert scale (“not at all true”, “not true”, “neither true”, “true” and “completely true”). Item C7 “Environmental pollution/environmental toxins” was relevant for the study as a cause attribution. Nine subscales were formed from the 64 items by summing up item clusters: Identity, Time Course (acute vs. chronic), Consequences, Personal Control, Treatment Control, Coherence, Cyclical Occurrence, Emotional Representation, and Causes.

### 2.4. NEO-FFI

The NEO-FFI is a multidimensional personality inventory that measures the dimensions of neuroticism, extraversion, openness to experience, empathy and conscientiousness. It has five scales and comprises a total of 60 items [11,12].

### 2.5. EQ-5D-Health-Related Quality of Life

The EQ-5D is a self-assessment instrument and describes the state of health on the basis of five dimensions [13]: Agility, mobility. Ability to take care of oneself. Everyday activities (e.g., work, study, housework, family, leisure). Pain, physical discomfort. Anxiety, depression. The dimensions are each asked with one item and have three possible answers (no, moderate and extreme problems). Using standardized calculation specifications, the answers to the five questions are converted into an index value that expresses the respondent’s state of health in a one-dimensional measure ranging from 0 (very poor) to 1 (best possible state of health). In the second part of the EQ-5D, patients rate their current state of health on a visual analog scale (VAS) between 0 (“worst conceivable state of health”) and 100 (“best conceivable state of health”). The VAS allows a general assessment of one’s own current state of health independently of the answers to the five dimensions of the questionnaire.

### 2.6. BSI—Brief Symptom Inventory

The BSI is a version of the Symptom Checklist 90 (SCL-90-R) by Derogatis [14] shortened to 53 items, a questionnaire for recording subjective impairment caused in particular by psychological but also physical symptoms. The items are assigned to nine scales (somatization, compulsiveness, insecurity in social contact, depressiveness, anxiety, aggressiveness/hostility, phobic anxiety, paranoid thinking, psychoticism) and allow the calculation of three indicators of global distress: Global Severity Index (GSI), Positive Symptom Distress Index (PSDI) and Positive Symptom Total (PST), whereby the GSI is considered the most sensitive indicator of the patient’s psychological distress. The PSDI records the intensity of distress, the PST comprises the number of items for which distress was reported. For the evaluation, the raw scores are transformed into T-values (T-GSI, T-PSDI, T-PST) using a norm table. T-scores between 40 and 60 cannot be considered clinically abnormal, as 2/3 of the norm group are in this range. In general, mental abnormality is assumed if the T-GSI is >63 or if the T-scores in at least two scales are greater than or equal to 63.

### 2.7. Study Procedure

The clinical trial consisted of two patient contacts: Initial contact (first presentation in the environmental medicine outpatient clinic) with an environmental medical history, physical examination and sampling for biomonitoring appropriate in relation to the medical history and including noxious substances suspected by the patient. Biosamples from urine and/or blood were taken for biomonitoring. This corresponded to the usual procedure in the outpatient clinic. The following psychometric tests are given to the patient at the first contact and completed by the patient in the waiting area: NEO-FFI, BSI, IPQ-R and EQ-5D.

Second contact was made after approx. 2–3 weeks to discuss the findings. The biomonitoring data were assessed in relation to established health-based guidance values, primarily those provided by the Human Biomonitoring Commission (HBM) of the German Environment Agency (Umweltbundesamt). Where available, HBM-I and HBM-II values served as reference points to contextualize individual results. In cases where no HBM values were available, orientation values or reference ranges from population studies were used. If at this point biomonitoring or other laboratory results were relevant to health and could explain the patient’s complaints, the patient was excluded from the study. In the case of inconspicuous findings, the patients were informed that the measured values were not elevated about guidance values and therefore not threatening to their health and that, fortunately, there was no poisoning; the complaints had a different cause than the suspected noxious substances that had been determined in the biomonitoring. We used this language as our primary aim was to provide reassurance based on available health-based guidance values and clinical toxicological assessments, as many patients presented with heightened anxiety and uncertainty. The study participants were then given the IPQ-R and EQ 5D scale psychometric questionnaires again and asked to complete them and return them by post to the environmental medicine outpatient clinic after 4 days at the earliest. This time interval was intended to give the patients the opportunity to deal with this result. The other tests (Neo-FF, BSI) were not repeated as no short-term change was expected. All patient contacts were carried out by the same outpatient clinic doctor (CS).

### 2.8. Case Number Planning and Statistics

Forty consecutive patients were planned for the study. With this number of cases, a change on the IPQ-R between the initial examination and the follow-up examination can be detected with an assumed effect size of d = 0.45 and a statistical power of 1 − ß = 0.8 (t-test for linked samples). For the further analyses, the participants were divided into two groups based on the information for item C7 of the IPQ-R (causes of illness; environmental pollution or environmental toxins): those who changed their illness model (“changed”) and those who did not change their illness model (“stable”). Based on this empirical division into two groups, the baseline data of the two groups were examined for differences. If the test subjects continued to attribute their symptoms to environmental toxins at time T2 (“true” and “completely true”), they were assigned to the “stable” group. Test subjects who no longer attributed their symptoms to environmental toxins at T2 (response options in item C7 of the IPQ-R: “not true at all”, “not true”, “neither”) were assigned to the “changed” group. The data of the two groups were checked for statistically significant differences using chi-square and *t*-tests, as well as repeated measures ANOVAS.

## 3. Results

Of 358 patients who visited the Environmental Medicine Outpatient Clinic at Freiburg between July 2013 and November 2015, 51 met the primary inclusion criteria and had no exclusion criteria. The most common reasons for exclusion were previously diagnosed MCS; sick-building syndrome or chronic fatigue syndrome.

Three patients (5.9%, molds *n* = 2, lead *n* = 1) were excluded due to health-relevant findings in the biomonitoring; in a further 15 patients there was no environmentally associated disease attribution in the IPQ-R at T1, so that they were not suitable for the primary research question. Therefore, 33 patients could be analyzed for the primary research question (Figure 1).

The baseline characteristics of the patients and the chemicals suspected by the patients are shown in Table 1.

### 3.1. IPQ-R

Out of the 33 patients, 7 (21.2%) patients changed their disease attribution measured with item C7 of the IQ-R after notification of the inconspicuous biomonitoring result; the other 26 patients did not change their disease attribution after notification of the inconspicuous biomonitoring result (Figure 2).

There were no differences between the original sample (*n* = 51) and the sample for the evaluation of the primary target parameter (*n* = 33) (Table 2).

Patients who changed their disease attribution after reporting normal findings during biomonitoring did not differ with regard to demographic data (gender, age, occupation). However, at T1 there was a significant difference between the two groups in the assessment of the consequences of the disease (t = −2.8, *p* < 0.01, Cohen’s d = −1.19) and the perception of personal control (t = −2.08, *p* < 0.05, Cohen’s d = −0.89). Patients who changed their disease attribution based on the biomonitoring results rated the consequences of the disease as less severe (14.0 ± 2.3) than those who did not change their disease attribution (17.3 ± 2.9). Prior to receiving biomonitoring results, the group of patients who changed their diseases attribution also felt that they had less personal influence on the disease (9.0 ± 2.5) than the group of patients who did not change their disease attribution (11.8 ± 3.2). In the ANOVA comparison with the patients who did not change their clinical picture (MT1 = 14.20, SDT1 = 1.87; MT2 = 14.40, SDT2 = 1.53), the patients who changed their clinical picture after the clarification felt that the illness would last longer than before the clarification (MT1 = 13.41, SDT1 = 0.72, MT2 = 15.36, SDT2 = 1.38). For details, see Appendix A.

### 3.2. NEO-FFI

The sample was within the normal range with regard to the personality traits measured using the NEO-FFI (Table 3). The personality traits did not differ between patients who changed their disease attribution and those who maintained their environmental attribution.

### 3.3. BSI T-Value Transformation

From the standardization table for adults (N = 600 m/f per 300), a T-GSI of 65 (n = 51) or 66 (n = 33), a T-PSDI of 65 or 64 and a T-PST of 59 or 61 result for our sample. Our sample can therefore be classified as conspicuous or mentally stressed.

Patients who changed their cause attribution had a trend towards a lower BSI total score (GSI, t = −2.02, *p* = 0.052, Appendix A); however, no significant differences could be observed. In the corresponding subscales (Figure 3, for more details see Appendix A), patients who changed their attribution showed a lower tendency to somatize, less compulsiveness, and a trend towards less depression.

### 3.4. EQ-5D

The EQ-5D index value for the age group of 45- to 54-year-olds corresponding to our sample is 0.945 from the standard population; the comparative value for the VAS is 78.517. The lower the values, the greater the stress. The corresponding values for our study population is 0.72 for the index and 51.04 for the VAS (N = 51). These figures suggest that patients referred for environmental health consultation are experiencing significant psychological distress. There was no difference between the two groups “environmental attribution changed” and “environmental attribution “stable” (Table 4).

## 4. Discussion

A key finding in our study is that only 21% of patients adjusted their disease attribution following inconspicuous biomonitoring results. This aligns with a similar interview-based study of 51 patients diagnosed with environmental sensitivities, in which 83% of those with a previous diagnosis of MCS or sick building syndrome continued to attribute their symptoms to “environmental” factors, despite only 8% of them having biologically verifiable exposures [17].

Interestingly, patients who maintained an environmental attribution perceived the consequences of their illness as more severe than those who revised their attribution. They also displayed a stronger internal locus of control, meaning they believed they had greater control over the course of their condition. Previous research has suggested that the combination of external cause attribution and internal locus of control can be beneficial for disease outcomes in conditions such as breast cancer [18] or post-accident recovery [19]. However, it remains uncertain whether this holds true when the environmental trigger is unverifiable. The type of disease attribution significantly influences quality of life, with some studies, such as those involving heart failure patients, showing that it may play a more important role in patient outcomes than objective clinical parameters [20].

BSI results demonstrate that psychological symptoms are significantly more prevalent among patients with environmental sensitivities. The evaluation according to the subscales of the BSI also showed significant differences in the “somatization” and “compulsivity” scales. The patients who adhered to their environmental attributes showed a more pronounced tendency towards somatization and compulsivity. However, the personality traits measured with the NEO-FFI did not significantly differ between the groups. In another study at a German Environmental Outpatient Clinic with 264 patients, 75% met the DSM-IV criteria for a psychiatric disorder and 35% suffered from somatoform disorders [21]. Further studies found rates of psychiatric morbidity in environmental medicine patients ranging from 36 to 100% [22,23]. However, the sample sizes are often small, and the psychiatric evaluation was not carried out using a standardized procedure. Interestingly, a recent prospective cohort study found after adjustment for chronic stress no significant alteration in heart rate variability, a parameter for the activity of the autonomous nervous system, in MCS patients [24].

The EQ-5D showed a clear disease burden in our sample. EQ-5D VAS values and corresponding index values for chronic diseases are, for example, 51 and 0.72 for psoriasis patients and 43 and 0.33 for cancer patients [25]. The burden of our sample corresponded approximately to that of psoriasis patients. The generalizability of the study results is limited by the relatively small number of cases. On the other hand, there were no dropouts, which is a strength of the study. We acknowledge that our study population consisted of patients who actively sought consultation at a University Outpatient Clinic for suspected environmental exposures. This self-selected, help-seeking group may differ significantly from individuals with comparable exposures who do not report symptoms or pursue medical evaluation. As such, a selection bias is likely present, particularly regarding patients’ pre-existing beliefs about environmental causation and their emotional investment in diagnostic outcomes. Our findings—especially those related to trust in biomonitoring and shifts in illness attribution—should therefore be interpreted in the context of this specific population. Future research could benefit from including comparison groups recruited from occupational settings with objectively documented chemical exposures but without prior symptom attribution.

Environmental patients often reject the explanation that the symptoms are psychosomatic or neurobiologically caused [26]. This observation is supported by the results of the present study. The immediate physical nature of symptoms triggered by chemical exposure often leads to dismissing psychological factors [27]. Consequently, the somatic disease model makes this patient group susceptible to unproven and often costly therapies. On the other hand, there are reports that the drop-out rate among patients who have accepted an offer of counseling is low [28].

Recently, a clinical study concluded that among various factors, unresolved emotional trauma contributing to both chronic and acute stress reactions might play a significant role in the development of MCS and can serve as a foundation for effective treatment [27]. There have been three distinct clinical presentations identified: the accidental type, which follows chemical exposure accompanied by an emotional trauma; the associative type, which arises from repeated low dose exposures in a threatening context; and the developmental type, which stems from traumatic experiences during childhood or adolescence, leading to hypervigilance and chronic stress or trauma-related disorders. Considering this, patients may benefit from exploring explanations beyond just environmental causes, allowing for a more comprehensive understanding and treatment approach.

Our study might contribute to the awareness that environmental complaints may stem from both environmental and specific psychological factors. This can lead to better communication, where physicians and actively listen to the emotional and mental health context of the patient’s experience. Broader understanding could encourage a more integrated treatment plan, addressing both physical symptoms and psychological conditions.

## 5. Conclusions

Targeted biomonitoring in “environmental patients” often yields inconspicuous results, with few altering their disease attribution afterward. Most maintain beliefs that environmental factors cause their symptoms, exhibiting higher somatization and compulsivity indices than those who revise their views. They perceive their illness as more severe yet more controllable through personal behavior.

## Figures and Tables

**Figure 1 ijerph-22-01143-f001:**
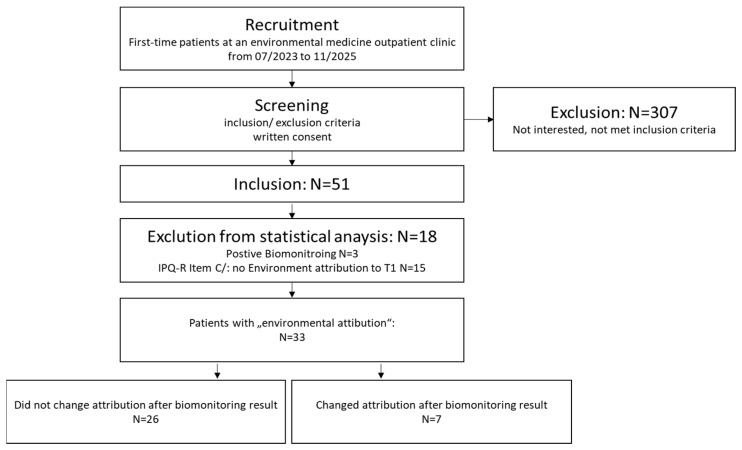
Flow chart for study participation.

**Figure 2 ijerph-22-01143-f002:**
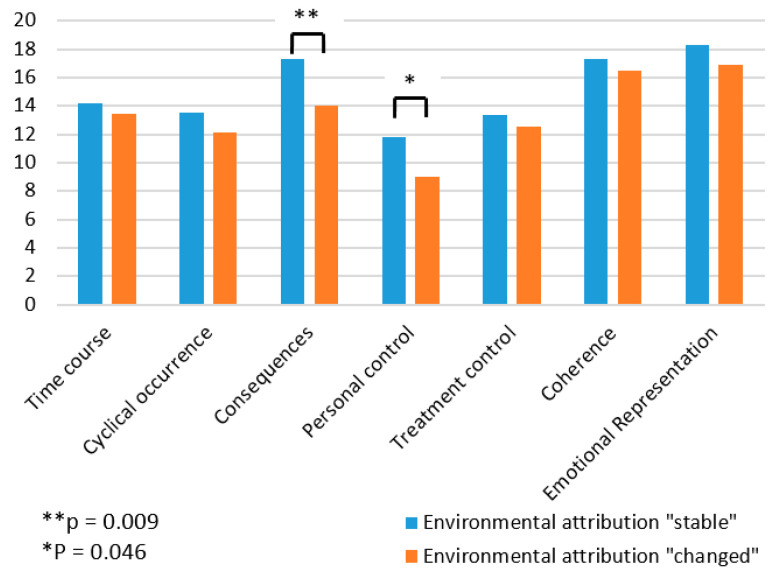
IPQ-R at T1.

**Figure 3 ijerph-22-01143-f003:**
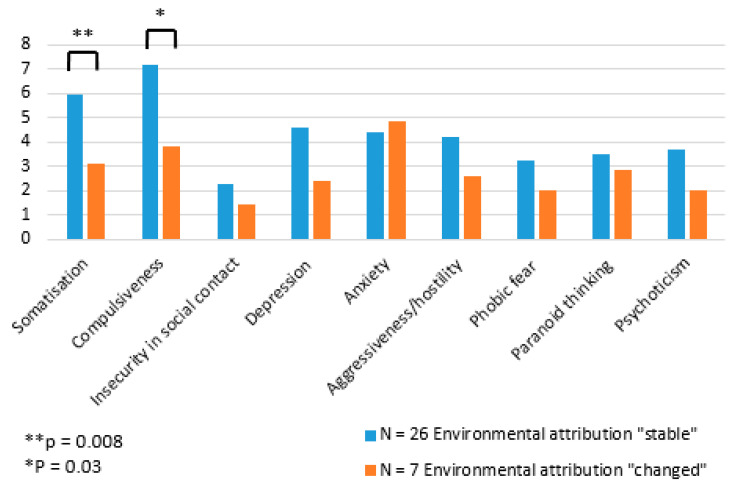
BSI subscales.

**Table 1 ijerph-22-01143-t001:** Sociodemographic and clinical characteristics at baseline.

Variable	*n* = 51	*n* = 33
Gender		
male	19 (37.3%)	13 (39.4%)
female	32 (62.7%)	20 (60.6%)
Age ^1^	50.8 ± 14.4	54.06 ± 13.56
Highest school-leaving qualification ^2^		
Secondary school leaving certificate	13 (27.1%)	7 (21.9%)
Secondary school leaving certificate	11 (22.9%)	9 (28.1%)
A-levels/Abitur	9 (18.8%)	5 (15.6%)
Completed university degree	15 (31.3%)	11 (34.4%)
Biomonitoring ^3^	Total (male/female)	Total (male/female)
Heavy metals not differentiated	18 (9/9)	14 (7/7)
Aluminum	1 (1/0)	1 (1/0)
Arsenic	1 (0/1)	1 (0/1)
Lead	8 (4/4)	4 (2/2)
Nickel	1 (0/1)	1 (0/1)
Mercury	10 (5/5)	7 (5/2)
Mold fungi	10 (4/6)	6 (2/4)
Wood preservatives	2 (2/0)	2 (2/0)
Occupational allergens (isocyanates, formaldehyde)	2 (0/2)	1 (0/1)
Organic solvents	4 (1/3)	2 (1/1)
PAH ^4^	4 (1/3)	3 (1/2)
Pesticide contamination	1 (0/1)	1 (0/1)
PCB ^5^	1 (1/0)	1 (1/0)

^1^ Mean ± standard deviation. ^2^ No information *n* = 3 of 51; *n* = 1 of 33 ^3^ Multiple answers possible ^4^ PAHs (polycyclic aromatic hydrocarbons) ^5^ PCBs (polychlorinated biphenyls).

**Table 2 ijerph-22-01143-t002:** BSI baseline values.

BSI	*n* = 51	*n* = 33	*n* = 26 Environmental Attribution “Stable”	*n*= 7 Environmental Attribution “Changed”	*p* “Stable” vs. “Changed”	d 95-%-CI
GSI	0.68 (0.37)	0.71 (0.35)	0.77 (0.35)	0.48 (0.24)	0.05	−0.875 (−1.735–−0.014)
T-GSI	65	66	68	58		
PSDI	1.61 (0.59)	1.57 (0.36)	1.61 (0.37)	1.43 (0.29)	0.24	−0.506 (−1.349–0.338)
T-PSDI	65	64	65	61		
PST	20.98 (10.23)	23.06 (9.30)	24.68 (9.50)	17.29 (5.99)	0.06	−0.828 (−1.686–0.031)
T-PST	59	61	63	55		

Mean values ± standard deviations, Global Severity Index (GSI), Positive Symptom Distress Index (PSDI) and Positive Symptom Total (PST), d = Cohen’s d for differences in mean values between groups of different sizes; CI, confidence interval.

**Table 3 ijerph-22-01143-t003:** NEO-FFI baseline values.

Neo-FFI (Range 0–48)	*n* = 51	*n* = 33	*n* = 26 Environmental Attribution “Stable”	*n* = 7 Environmental Attribution “Changed”	*p* “Stable” vs. “Changed”	d 95-%-CI
Neuroticism	21.53 (7.23)	22.52 (7.71)	23.46 (7.67)	19.00 (7.28)	0.18	−0.587 (−1.434–0.259)
Extraversion	26.19 (7.24)	25.22 (7.42)	24.31 (7.79)	28.57 (4.96)	0.18	0.581 (−0.265–1.428)
Openness to experience	30.09 (6.29)	29.21 (6.31)	28.77 (6.42)	30.86 (6.07)	0.45	0.329 (−0.509–1.167)
Agreeableness	31.61 (4.87)	31.12 (4.92)	30.46 (5.14)	33.57 (3.21)	0.14	0.644 (−0.205–1.493)
Conscientiousness	34.27 (5.64)	33.49 (5.47)	33.51 (5.84)	33.43 (4.24)	0.97	−0.014 (−0.849–0.82)

Mean values ± standard deviations, d = Cohen’s d for mean differences between groups of different sizes; CI, confidence interval.

**Table 4 ijerph-22-01143-t004:** EQ 5D baseline values.

EQ-5D	*n* = 51	*n* = 33	*n* = 26 Environmental Attribution “Stable”	*n* = 7 Environmental Attribution “Changed”	*p* “Stable” vs. “Changed”	d 95-%-CI
Health index	0.72 (0.28)	0.76 (0.23)	0.75 (0.24)	0.79 (0.22)	0.62	0.169 (−0.66–1.005)
VAS (“thermometer”)	51.04 (24.30)	54.2 (23.13)	52.23 (24.14)	61.43 (18.64)	0.36	0.397 (−0.4–1.2)

Mean values ± standard deviations; d = Cohen’s d for mean differences between groups of different sizes; CI, confidence interval.

## Data Availability

The datasets used and/or analyzed during the current study are available from the corresponding author on reasonable request.

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
