# Peer review of "Do Patients with Complaints Attributed to Chemicals in the Environment Trust in Biomonitoring as a Valid Diagnostic Tool? A Prospective, Observational Study from a German University Outpatient Clinic"

_ijerph, 2025, doi:10.3390/ijerph22071143_

Round 1

Reviewer 1 Report

Comments and Suggestions for Authors

The starting point of this research is very valuable: in particular, (how) can human biomonitoring be a useful tool in the practice of an environmental health consultation? In some cases this may reveal a cause for real health complaints, in other cases it may not reveal much, but it can be an important part of a broader search by patients, including psychological help. However, the paper contains too little information to fully appreciate the broader context and the way in which biomonitoring is used (e.g. how exactly do patients end up at the outpatient clinic? What are the starting points for the conversation? What is the role of biomonitoring in this context? How are the biomonitoring results assessed? Is there openness to other possible interpretations and uncertainties? ...) The limited effectiveness of biomonitoring in this context may also be a consequence of the way in which it is used?

In general, the text is not very nuanced and lacks conceptual clarity and consistency. The introduction and interpretation are only partially elaborated and it would be good to better situate within the scientific literature. At times the text lacks some empathy for the target group, which seems to consist at least partly of patients who primarily need psychological help.

A more concrete list of points for improvement follows below.

It is important to emphasize that I do not have a medical background. I have a background in environmental sciences and sociology, and my expertise focuses on the use of human biomonitoring in a general population, usually in the context of specific environmental problems. In that context, I study, among other things, the impact of human biomonitoring on risk perception and effective strategies for risk communication and governance. My review should therefore be read from that point of view.

(Specific) points for improvement of the paper:

The article is difficult to read and at times unclearly structured. For example, the introduction does not follow a clear storyline. The explanation about multiple chemical sensitivity (MCS) in the introduction (lines 44 to 71) comes out of the blue and is insufficiently framed.

I am not a medical expert, but a more nuanced description of MCS (and also sick building syndrome) seems appropriate. Due to the lack of clarity about this when reading the article, I consulted other sources and read for example that there is still much discussion about the exact causes, be it medical or rather psychological, as well as diagnosis, symptoms and responses.

A bit more explanation about the ‘environmental medicine outpatient clinic’ would be useful. How exactly do patients end up here? For example, by referral or on their own initiative? For which type of expertise? This would help to better situate the context of the research and the composition of the participant group. (Further on in the text I read that there are also patients in the participant group with a lack of environmental attribution of complaints. So the outpatient clinic is not only for patients who suspect an environmental cause for complaints?)

The reasoning behind the exclusion criteria is not clear. Why, for example, are patients with concomitant diseases, and in particular previously diagnosed MCS or sick building syndrome, excluded? This seems to be the reason why most patients come to the outpatient clinic and this research is intended to investigate whether biomonitoring can be a trusted tool in that context? Why is a distinction made between patients who have been diagnosed before and patients who have not been diagnosed (yet)?

Study Procedure:

  • Line 168: what is meant by ‘environmental medical history’?
  • What do the participants expect from the first consultation? Do they know in advance that a sample will be taken for biomonitoring? Do they come specifically for that? Or is this part of a broader consultation in which also other medical hypotheses are being investigated?
  • Line 174: ‘a measured value’ – in the biomonitoring?
  • The text does not explain how the biomonitoring data are being assessed in terms of health. Are health-based guidance values used? If so, which ones? Is there openness to other interpretations and uncertainties? Or limitations of the measurement methods? …
  • Line 177: Words like 'harmless' and ‘no poisoning’ suggest that chemical exposure is viewed rather in black and white terms, while there is often scientific discussion about this. For example, i) the WHO states that there is no safe exposure level for lead and other carcinogenic substances, ii) Health based guidance values ​​evolve regularly, iii) we are not exposed to single, but to a cocktail of chemicals. This type of nuances are missing from the text.
  • Line 181: 4 days can be quite short for the necessary reflection? Was there much variation in this interval among participants? And would that have an influence on the answers?

Results:

  • Table 2: What is the meaning of the numbers behind the pollutants in the 'biomonitoring' section?
  • Lines 254-260: what is meant by ‘the clarification’? and ‘their clinical picture’? A more precise formulation seems appropriate.
  • Line 264: ‘attitude’ -> perception? (attitude and perception are related, but not synonyms.)
  • Line 275: ‘obsessiveness’ -> compulsiveness?
  • Lines 283-284: ‘our environmental patients’ -> This wording was not previously used in the text. It is best to remain consistent in terminology. This also applies to other concepts in the paper.
  • Line 284: ‘considerably stressed’ -> a more nuanced wording seems appropriate.

Discussion:

  • Lines 293-296: ‘environmental sensitivities’ and ‘verifiable environmental exposures’: these wordings were not previously used in the text, however these concepts sound more solid and nuanced compared to other concepts used in earlier parts of the paper. So I would advise to use these concepts in a consistent way throughout the paper.
  • Lines 308-309: this analysis sounds rather unempathetic.
  • Line 313: what is meant by ‘unremarkable’?
  • Line 343: ‘intoxication’ -> see previous comments. A bit more nuance in the wording would be desirable.
Comments on the Quality of English Language

I am not a language expert myself, but I think a language review would be useful. The text also lacks consistency in the concepts used.

The text contains some strange phrases:

  • Line 72: ‘alleged exposure to environmental medicine’ ?
  • Line 81: ‘whether the exclusion of noxious substances suspected by the patient in biomonitoring has an influence on’ ?
  • Line 84: ‘the targeted exclusion of environmental medical exposure’ ?
  • Line 95: ‘suspected measurable, environmentally associated complaints’
  • Line 241 and others: ‘collective’ -> sample?

Author Response

Open Review I

The starting point of this research is very valuable: in particular, (how) can human biomonitoring be a useful tool in the practice of an environmental health consultation? In some cases this may reveal a cause for real health complaints, in other cases it may not reveal much, but it can be an important part of a broader search by patients, including psychological help. However, the paper contains too little information to fully appreciate the broader context and the way in which biomonitoring is used (e.g. how exactly do patients end up at the outpatient clinic? What are the starting points for the conversation? What is the role of biomonitoring in this context? How are the biomonitoring results assessed? Is there openness to other possible interpretations and uncertainties? ...) The limited effectiveness of biomonitoring in this context may also be a consequence of the way in which it is used?

Thank you for this insightful comment. We expanded the Methods and Discussion to clarify how biomonitoring is embedded in our clinical workflow. The environmental medicine outpatient clinic is staffed by physicians with specialized training in environmental health, clinical toxicology and exposure assessment. Patients may be referred by general practitioners or specialists—or they may self‑refer—when symptoms are suspected to involve environmental factors, but the service also welcomes individuals with unexplained complaints regardless of prior attribution. This multi-disciplinary setting thus serves both those who arrive expecting an environmental diagnosis and those seeking a broader differential evaluation; at the first visit we take an open‑ended history to elicit each patient’s own exposure narrative, which then guides selection of the analytes. Sampling for biomonitoring was appropriate in relation to the medical history and always included the noxious substances suspected by the patient. Laboratory results are interpreted against HBM I/II and other national reference values. In the consultation, we present these findings in lay terms and explicitly invite consideration of alternative explanatory models—biopsychosocial—documenting these discussions qualitatively.

The article is difficult to read and at times unclearly structured. For example, the introduction does not follow a clear storyline. The explanation about multiple chemical sensitivity (MCS) in the introduction (lines 44 to 71) comes out of the blue and is insufficiently framed.

We revised the introduction to ensure a clear storyline and with regard to other reviewers comments.

I am not a medical expert, but a more nuanced description of MCS (and also sick building syndrome) seems appropriate. Due to the lack of clarity about this when reading the article, I consulted other sources and read for example that there is still much discussion about the exact causes, be it medical or rather psychological, as well as diagnosis, symptoms and responses.

We have revised the introduction: Two related but distinct syndromes are commonly discussed in this context: multiple chemical sensitivity (MCS ICD‑10 T78.4) and sick building syndrome (SBS). MCS is defined by chronic, reproducible symptoms triggered by low‑dose exposure to chemically diverse agents—responses not explained by classical toxicology or allergy—and involving multiple organ systems. In contrast, SBS describes symptoms (e.g., headache, mucosal irritation, fatigue) that are temporally associated with time spent in a specific building, often improving upon leaving that environment, and usually occurring in several individu-als exposed to the building.

A bit more explanation about the ‘environmental medicine outpatient clinic’ would be useful. How exactly do patients end up here? For example, by referral or on their own initiative? For which type of expertise? This would help to better situate the context of the research and the composition of the participant group. (Further on in the text I read that there are also patients in the participant group with a lack of environmental attribution of complaints. So the outpatient clinic is not only for patients who suspect an environmental cause for complaints?)

The environmental medicine outpatient clinic is staffed by physicians with specialized training in environmental health, clinical toxicology and exposure assessment. Patients may be referred by general practitioners or specialists—or they may self‑refer—when symptoms are suspected to involve environmental factors, but the service also welcomes individuals with unexplained complaints regardless of prior attribution. This multidisciplinary setting thus serves both those who arrive expecting an environmental diagnosis and those seeking a broader differential evaluation.

The reasoning behind the exclusion criteria is not clear. Why, for example, are patients with concomitant diseases, and in particular previously diagnosed MCS or sick building syndrome, excluded? This seems to be the reason why most patients come to the outpatient clinic and this research is intended to investigate whether biomonitoring can be a trusted tool in that context? Why is a distinction made between patients who have been diagnosed before and patients who have not been diagnosed (yet)?

We excluded patients with significant comorbidities or established diagnoses (e.g., MCS, sick building syndrome) to ensure that our cohort reflected individuals at their very first presentation, without prior condition‑specific interventions or counseling. This “clean” baseline reduces confounding from other diseases or treatments and prevents expectation bias arising from a pre‑existing diagnostic framework. Consequently, we can more accurately assess how incorporating biomonitoring data influences initial clinical reasoning and management decisions. We added this.

    Line 168: what is meant by ‘environmental medical history’?

An environmental medical history is a focused patient interview that records exposures to potential hazards in one’s home, workplace, hobbies, diet and lifestyle (e.g., chemicals, mold, dust, consumer products). It links symptom patterns (onset/times/locations when they improve or worsen) to these exposures to guide further testing or interventions.

    What do the participants expect from the first consultation? Do they know in advance that a sample will be taken for biomonitoring? Do they come specifically for that? Or is this part of a broader consultation in which also other medical hypotheses are being investigated?

In most cases, participants come to the first consultation with the expectation that biomonitoring will be conducted. They usually present with symptoms that they associate with environmental exposures, or bring environmental samples (e.g., dust, materials, or air measurements) for discussion. The consultation itself is embedded in a broader clinical approach that includes a detailed medical and exposure history, differential diagnostic considerations, and, where appropriate, the initiation of biomonitoring. Thus, while many patients specifically seek an evaluation of chemical exposure, the consultation also explores other possible medical explanations in line with an integrative and environmental medicine framework.

    Line 174: ‘a measured value’ – in the biomonitoring?

This refers to all laboratory exams including biomonitoring, we clarified this.

    The text does not explain how the biomonitoring data are being assessed in terms of health. Are health-based guidance values used? If so, which ones? Is there openness to other interpretations and uncertainties? Or limitations of the measurement methods? …

The biomonitoring data were assessed in relation to established health-based guidance values, primarily those provided by the Human Biomonitoring Commission (HBM) of the German Environment Agency (Umweltbundesamt). Where available, HBM-I and HBM-II values served as reference points to contextualize individual results. In cases where no HBM values were available, orientation values or reference ranges from population studies were used. We added this to the study procedure.

We are aware that some of these reference values may be outdated, and that HBM values are lacking for many substances. However, to support patients’ sense of safety and to avoid generating unnecessary concern, we did not emphasize these uncertainties in the consultation. General limitations of biomonitoring, such as interindividual variability and the potential effects of chemical mixtures, were addressed where appropriate to provide a transparent and differentiated view.

    Line 177: Words like 'harmless' and ‘no poisoning’ suggest that chemical exposure is viewed rather in black and white terms, while there is often scientific discussion about this. For example, i) the WHO states that there is no safe exposure level for lead and other carcinogenic substances, ii) Health based guidance values ​​evolve regularly, iii) we are not exposed to single, but to a cocktail of chemicals. This type of nuances are missing from the text.

We appreciate this important observation. Indeed, the scientific discourse on chemical exposure is complex and evolving, particularly regarding cumulative effects, low-dose exposures, and the absence of clear thresholds for certain substances. In our text, the terms such as “harmless” or “no poisoning” were used in the context of patient communication. Our primary aim was to provide reassurance based on available health-based guidance values and clinical toxicological assessments, as many patients presented with heightened anxiety and uncertainty. We now avoided these expressions, or clarified in the revised manuscript that these expressions were used in a therapeutic context and do not imply a simplistic or binary view of chemical exposure from a scientific perspective.

    Line 181: 4 days can be quite short for the necessary reflection? Was there much variation in this interval among participants? And would that have an influence on the answers?

We defined a minimum interval of four days between the discussion of biomonitoring results and the completion of the questionnaires to ensure sufficient time for initial reflection. While we did not systematically record the exact duration for each participant, we cannot assess the extent of variation in reflection time or its potential influence on the responses. Informal feedback suggested that this timeframe was generally perceived as adequate.

    Table 2: What is the meaning of the numbers behind the pollutants in the 'biomonitoring' section?

In Table 1, the numbers behind the pollutants indicate the number of patients that attibute the symptoms to the particular pollutant. Total number (male/female). We have indicated this more clearly now.

    Lines 254-260: what is meant by ‘the clarification’? and ‘their clinical picture’? A more precise formulation seems appropriate.

We changed the sentence to: Prior to receiving biomonitoring results, the group of patients who changed their diseases attribution also felt that they had less personal influence on the disease (9.0 ± 2.5) than the group of patients who did not change their disease attribution (11.8 ± 3.2).

    Line 264: ‘attitude’ -> perception? (attitude and perception are related, but not synonyms.)

We changed attitude to disease attribution

    Line 275: ‘obsessiveness’ -> compulsiveness?

We changed obsessiveness to compulsiveness

    Lines 283-284: ‘our environmental patients’ -> This wording was not previously used in the text. It is best to remain consistent in terminology. This also applies to other concepts in the paper.

We changed the phrase to „study population“.

    Line 284: ‘considerably stressed’ -> a more nuanced wording seems appropriate.

We updated to “experiencing significant psychological distress” to convey a more nuanced assessment of the patients’ psychosocial burden.

    Lines 293-296: ‘environmental sensitivities’ and ‘verifiable environmental exposures’: these wordings were not previously used in the text, however these concepts sound more solid and nuanced compared to other concepts used in earlier parts of the paper. So I would advise to use these concepts in a consistent way throughout the paper.

We have used the term “environmental sensitivities” (to describe self‑reported symptomatology linked to environmental agents) when possible, however the term „multiple chemical sensitivity“ is the defined term of the diasese.  We have also standardized “biologically verifiable exposures” (to refer to exposures measured by biomonitoring) throughout the text. All instances of alternative phrasing (e.g. “suspected exposures,” “objective evidence of exposure”) have been revised for consistency.

    Lines 308-309: this analysis sounds rather unempathetic.

We have changed it to: BSI results demonstrate that psychological symptoms are significantly more prevalent among patients with environmental sensitivities.

    Line 313: what is meant by ‘unremarkable’?

We have changed the sentence to: However, the personality traits measured with the NEO-FFI did not significantly differ between the groups.

    Line 343: ‘intoxication’ -> see previous comments. A bit more nuance in the wording would be desirable.

We have changed it to: environmental causes

    Line 72: ‘alleged exposure to environmental medicine’ ?

We have changed the phrase to: suspected environmental sensitivities

    Line 81: ‘whether the exclusion of noxious substances suspected by the patient in biomonitoring has an influence on’ ?

We have changed the sentence to: In the present study, we investigated whether the exclusion of biologially verifiable exposures by biomonitoring has an influence on the patients‘ disease model and quality of life.

    Line 84: ‘the targeted exclusion of environmental medical exposure’ ?

We have changed the sentence to: Drawing on our experience at the Environmental Medicine Outpatient Clinic of Freiburg, we identified two distinct patient response profiles to biomonitoring: those whose anxiety is relieved by the targeted exclusion of biologically verifiable exposures—leading them to abandon environmental noxae as the cause of their symptoms (the “changed” group)—and those whose personal disease attibution remains unchanged despite inconspicuous biomonitoring results (the “stable” group).

    Line 95: ‘suspected measurable, environmentally associated complaints’

We have changed the sentence to: We included patients aged 18 and older who, on their first visit to our university’s Environmental Medicine Outpatient Clinic, presented with complaints suspected to arise from quantifiable environmental exposures—specifically heavy metals, wood preservatives, pesticides, organic solvents, or mold spores.

    Line 241 and others: ‘collective’ -> sample?

We have replaced the term „collective“ with „sample“ throughout the manuscript.

Reviewer 2 Report

Comments and Suggestions for Authors

Overall, the subject is carefully chosen, and this work has the potential to significantly advance environmental health research.

The title is explicit and answers a particular problem about patient perception and environmental health. However, if it's geographically restricted, you might think about defining the population (for example, "in Germany" or "among general practitioners' patients").

Abstract is clearly written.

Rewrite introduction adding literature from relevant studies. Most of the cited references are irrelevant. 

A thorough description of the study population and methodology, precise definitions of important words (such as biomonitoring and trust), and a careful interpretation of the results in light of prior research and recognized biases are all necessary to strengthen the work.

Study design and presentation should carefully consider.

On analytical side why not authors consider biological samples of the patients like urine and blood?

Comments on the Quality of English Language

OK

Author Response

Open Review II

The title is explicit and answers a particular problem about patient perception and environmental health. However, if it's geographically restricted, you might think about defining the population (for example, "in Germany" or "among general practitioners' patients").

We have added to the title: A prospective, observational cohort from a German university outpatient clinic

Rewrite introduction adding literature from relevant studies. Most of the cited references are irrelevant.

We have thoroughly reviewed and revised the Introduction that better reflect the current state of research in biomonitoring, Multiple Chemical Sensitivity (MCS), and patient risk perception in environmental medicine.

A thorough description of the study population and methodology, precise definitions of important words (such as biomonitoring and trust), and a careful interpretation of the results in light of prior research and recognized biases are all necessary to strengthen the work.

We have expanded the manuscript to provide a detailed description of the study population and methodology, ensuring transparency and reproducibility. Key terms such as “biomonitoring” and “trust” are now clearly defined based on current literature to establish a common understanding. Additionally, we have deepened the interpretation of our findings by situating them within the context of prior research and addressing potential biases and limitations.

Study design and presentation should carefully consider.

We have clarified and strengthened the presentation of the study design throughout the manuscript. The revised Methods section now clearly outlines the prospective observational design, inclusion criteria, timing of assessments (pre- and post-biomonitoring feedback), and the rationale for grouping patients based on changes in illness attribution. We also provide a transparent description of the biomonitoring procedures, the use of standardized questionnaires, and our mixed-methods approach. In the revised Discussion, we further reflect on how the design supports our research objectives, while acknowledging methodological limitations and potential sources of bias. These changes aim to enhance both the clarity and credibility of the study’s structure and findings.

On analytical side why not authors consider biological samples of the patients like urine and blood?

Our biomonitoring approach primarily utilizes biological samples such as blood and urine to assess internal chemical exposures. The selection of specific matrices depends on the target analytes and their validated biomarkers—for example, blood lead levels or urinary metabolites of solvents. We have clarified this point in the study procedure section to avoid any ambiguity and to emphasize that biological sampling is central to our assessment of individual exposure.

Reviewer 3 Report

Comments and Suggestions for Authors

Dear Authors:

I have read your manuscript with interest, particularly the methodological approach and the psychometric evaluation of illness perception before and after biomonitoring. The study offers relevant insights into how patients interpret environmental exposures and support or adjust their illness attribution.

I would like to share one methodological observation that may be worth considering. Given that your sample was composed of patients who sought medical attention specifically for suspected environmental exposures, it is possible that a selection bias influenced the results. It could be useful to reflect on how your findings might differ if participants were recruited through random sampling in occupational settings where exposure to mixtures of xenobiotics is objectively present but not necessarily self-attributed.

Including this consideration in the discussion may help clarify the scope and applicability of your conclusions.

Author Response

Open Review III

I would like to share one methodological observation that may be worth considering. Given that your sample was composed of patients who sought medical attention specifically for suspected environmental exposures, it is possible that a selection bias influenced the results. It could be useful to reflect on how your findings might differ if participants were recruited through random sampling in occupational settings where exposure to mixtures of xenobiotics is objectively present but not necessarily self-attributed. Including this consideration in the discussion may help clarify the scope and applicability of your conclusions.

We agree that the self-selected nature of our sample—patients who actively sought consultation for suspected environmental exposures—may introduce a selection bias, particularly with regard to illness attribution and sensitivity to biomonitoring feedback. We have addressed this point in the revised Discussion section by noting that our findings reflect the perceptions and responses of a help-seeking population with pre-existing concerns, which may not be generalizable to individuals with similar exposures in occupational settings who do not report symptoms or seek care.

We now explicitly state that recruiting participants from occupational cohorts with objectively documented exposure—but without self-attribution—could yield different results regarding trust in biomonitoring, emotional response, and illness narrative. We agree that such comparative studies would be a valuable next step to better delineate the influence of subjective attribution versus actual exposure levels. This addition helps clarify the scope and limits of our conclusions.

Round 2

Reviewer 1 Report

Comments and Suggestions for Authors

Thank you for this revised version. My comments were adequately addressed.

From my experience with HBM in population research, we use a more open communication strategy, with more attention to uncertainties, knowledge gaps and potential diversity of interpretations, focused on transparency and offering a perspective for (precautionary) action. However, I do understand that there are other points of attention in this diagnostic setting. In that respect, I am glad that I was allowed to review the article.